# A tailored approach towards informing relatives at risk of inherited cardiac conditions: study protocol for a randomised controlled trial

Lieke M van den Heuvel,[1] Yvonne M Hoedemaekers,[2] Annette F Baas,[3] J Peter van Tintelen,[1] Ellen M A Smets,[4] Imke Christiaans[1]

**To cite:** van den Heuvel LM, Hoedemaekers YM, Baas AF, *et al*. A tailored approach towards informing relatives at risk of inherited cardiac conditions: study protocol for a randomised controlled trial. *BMJ Open* 2019;**9**:e025660. doi:10.1136/bmjopen-2018-025660

[1]Department of Clinical Genetics, Amsterdam University Medical Centres, Amsterdam, The Netherlands
[2]Department of Clinical Genetics, University Medical Centre Groningen, Groningen, The Netherlands
[3]Department of Genetics, University Medical Center Utrecht, Utrecht, The Netherlands
[4]Department of Medical Psychology, Amsterdam University Medical Centres, Amsterdam, The Netherlands

**Correspondence to**
Dr Imke Christiaans;
i.christiaans@amc.nl

## ABSTRACT

**Introduction** In current practice, probands are asked to inform relatives about the possibility of predictive DNA testing when a pathogenic variant causing an inherited cardiac condition (ICC) is identified. Previous research on the uptake of genetic counselling and predictive DNA testing in relatives suggests that not all relatives are sufficiently informed. We developed a randomised controlled trial to evaluate the effectiveness of a tailored approach in which probands decide together with the genetic counsellor which relatives they inform themselves and which relatives they prefer to have informed by the genetic counsellor. Here, we present the study protocol of this randomised controlled trial.

**Methods** A multicentre randomised controlled trial with parallel-group design will be conducted in which an intervention group receiving the tailored approach will be compared with a control group receiving usual care. Adult probands diagnosed with an ICC in whom a likely pathogenic or pathogenic variant is identified will be randomly assigned to the intervention or control group (total sample: n=85 probands). Primary outcomes are uptake of genetic counselling and predictive DNA testing by relatives (total sample: n=340 relatives). Secondary outcomes are appreciation of the approach used and impact on familial and psychological functioning, which will be assessed using questionnaires. Relatives who attend genetic counselling will be asked to fill out a questionnaire as well.

**Ethics and dissemination** Ethical approval was obtained from the Medical Ethical Committee of the Amsterdam University Medical Centres (MEC 2017-145), the Netherlands. All participants will provide informed consent prior to participation in the study. Results of the study on primary and secondary outcome measures will be published in peer-reviewed journals.

**Trial registration number** NTR6657; Pre-results.

## INTRODUCTION

Inherited cardiac conditions (ICCs) such as cardiomyopathies and primary arrhythmia syndromes generally demonstrate an autosomal dominant inheritance pattern and a wide variety of symptoms that can manifest

### Strengths and limitations of this study

► This randomised controlled trial investigates both the uptake of genetic counselling and of predictive DNA testing, as well as the acceptance and impact on psychological and family functioning in the tailored versus the standard approach, in probands and relatives.
► This study will be conducted in three clinical genetics clinics with expertise on cardiogenetics, which will facilitate participant inclusion.
► In this trial, evaluation of the effect on outcome of the different components of the intervention is not possible, due to limited power.
► In this randomised controlled trial it is not possible to blind participants, genetic counsellors or the executing investigator for the chosen intervention.
► Because a baseline measure for the secondary outcomes is not possible, we cannot control for likely confounding factors such as intention to inform at-risk relatives, and family and psychological functioning at baseline.

at any age.[1 2] One feared outcome is sudden cardiac death (SCD), which can occur at a young age and be the first symptom of disease.[3 4] With an incomplete penetrance and high variability in expression even within families, carriers of a familial variant may remain undetected but still be at risk for SCD even though treatment options are available that prevent disease progression or potentially life-threatening arrhythmias.[5] Predictive DNA testing is therefore offered to first-degree relatives of probands (the first person in a family diagnosed with an ICC) in whom a pathogenic variant is identified because these relatives are at 50% risk of also having inheriting the genetic variant.[5 6] Predictive DNA testing is offered to relatives in a stepwise manner (cascade screening), with the aim of identifying asymptomatic carriers of the familial variant to facilitate timely treatment.

Non-carriers of the familial variant generally do not need cardiac monitoring and can be reassured about their own risk and that of their offspring.[6]

In current practice in the Netherlands, probands are asked to inform their relatives, supported by a family letter written by the genetic counsellor. This is referred to as the family mediated approach.[7] Previous research, however, shows that uptake (the number of relatives at risk attending genetic counselling and/or undergoing predictive DNA testing) is relatively low in ICCs, particularly for cardiomyopathies. Reported uptakes are around 50% despite family letters being provided to a majority of relatives by the proband.[8–10] Previous research in other genetic patient populations, such as hereditary types of cancer, shows similar uptake percentages.[11–13]

Some relatives who do not attend genetic counselling will have deliberately decided against predictive DNA testing. However, the low uptake percentages also suggest that many relatives may be unaware, or insufficiently aware, of the risks involved and/or the possibilities for genetic counselling and subsequent surveillance and treatment. This is supported by research on family communication in ICCs. Patients are not always able to inform or correctly inform their relatives for a number of reasons, including disengagement with relatives, lack of understanding of the importance of the information, preoccupation with their own grief, difficulties in conveying the complex information to relatives or a wish to prevent burdening relatives by informing them about genetic risks.[8 14–18]

Previous studies assessing interventions to enhance family communication in hereditary diseases showed that some interventions are effective in increasing the uptake of genetic counselling.[19–21] An intervention trial aimed at improving family communication in specifically dilated cardiomyopathy is still ongoing.[22] A few studies have been published on more active approaches to informing relatives at risk in which healthcare professionals (HCPs) contact at-risk relatives directly.[23–26] These studies suggest that a more active approach can almost double the uptake of genetic counselling and predictive DNA testing by relatives. However, some of these studies were performed in a research setting (eg, in relatives already registered in research databases for the genetic disease), hampering direct translation of these results to a diagnostic setting. To our knowledge, more active approaches in patients with ICCs have not been studied thus far. However, a study by Ormondroyd et al[14] suggests that relatives eligible for predictive DNA testing for hypertrophic cardiomyopathy and long QT syndrome would support a more active approach to informing relatives at risk.

Although studies on more active approaches did not report any psychological harm in relatives at group level, these approaches could cause more unwarranted worry or pressure on relatives to opt for predictive DNA testing.[23–25] An active approach to informing relatives at risk could also breach the autonomy and confidentiality of probands, and may harm relative's right not to know.[27–29]

Furthermore, HCPs are often unaware of interpersonal dynamics within families and the personal circumstances of relatives at risk. Active approaches may therefore have a negative impact on family relationships or may cause psychological distress in both probands and relatives.[30]

Because of this, a tailored approach in which a proband decides together with the genetic counsellor which at-risk relatives he or she will inform and which relatives he or she prefers to be informed by the genetic counsellor could be optimal. With this approach, the probands expert knowledge of a relative's functioning and of family dynamics could be used appropriately, and the autonomy of the proband preserved. At the same time, more relatives at risk would be sufficiently informed.[28 30] Furthermore, probands for whom informing relatives is difficult or burdensome might be relieved or supported by this approach.[30]

## Objectives

The primary aim of this randomised controlled trial is to assess whether uptake of genetic counselling and testing of relatives at risk of an ICC will be increased by using a tailored approach to information provision for relatives, instead of usual care (ie, the family-mediated approach). Secondary objectives are to evaluate how such a tailored approach is appreciated by both probands and relatives as compared with usual care. In addition, this study aims to assess the perceived impact on family relationships and psychological functioning of both probands and relatives. The protocol presented here has been described based on the 'Standard Protocol Items: Recommendations for Interventional Trials (SPIRIT) statement.[31]

## METHODS

### Design

A multicentre randomised controlled trial with a parallel-group design will be conducted in three university hospitals in the Netherlands (the Amsterdam University Medical Centres (Amsterdam UMC), the University Medical Centre Utrecht (UMCU) and the University Medical Centre Groningen (UMCG)) to compare the effects of a tailored approach to informing relatives at risk of ICCs to usual care in both probands and relatives.

### Participants

All probands aged 18 years or older with an ICC, or suspicion thereof, attending pre-test genetic counselling at the cardiogenetics outpatient clinics during the inclusion period will be asked to participate if they: (1) are the first member of their family to visit the cardiogenetics outpatient clinic for counselling about genetic testing for ICCs; (2) have at least one living adult relative; and (3) are able to read and write Dutch. Only probands in whom a likely pathogenic or pathogenic variant is detected (class 4—likely pathogenic or class 5—pathogenic variant) will be definitively included.

In addition, eligible adult first/second-degree relatives of enrolled probands who make an appointment

at the cardiogenetics outpatient clinics will be invited to fill out a questionnaire to measure secondary outcomes. Inclusion criteria are defined as follows: (1) first-degree adult (18 years or older) relatives of probands enrolled in the study or second-degree adult relatives in case of a deceased connecting first-degree relative who was affected or suspected to be affected, and (2) able to read and write Dutch.

## Procedure
Figure 1 shows a flowchart of the study procedure.

### Recruitment and consent
During pre-test genetic counselling, the genetic counsellor will inform the probands about the study and provide an informational letter (see online supplementary material S1). In addition, probands will be asked if the executing researcher can contact them to provide further information about the study. Subsequently, probands will be contacted by telephone by the executing researcher. If probands are still interested in participation, written informed consent forms will be sent by post, including a return envelope. As described earlier, only probands in whom a likely pathogenic or pathogenic variant is detected will be definitively included in the study.

Relatives of enrolled probands attending pre-test genetic counselling in one of the participating centres who are also at risk will also be invited to participate in the study. The same recruitment procedure will be used.

### Randomisation
Prior to receiving their test result, probands with an ICC in whom a likely pathogenic or pathogenic variant is identified will be randomly assigned to either the intervention or control group. Block randomisation will be used, with variable blocks ranging from size two to six. Randomisation will be stratified for gender, disease type (cardiomyopathies or primary arrhythmia syndromes) and hospital. To ensure allocation concealment, computer software will be used for randomisation, with an allocation rate of 1:1.[32] Relatives of probands included in the study will be assigned to the group to which the proband was assigned.

Neither participants nor genetic counsellors will or can be blinded for group assignment. The executing researcher also cannot be blinded because of slight differences between the questionnaires administered in the intervention and control groups. Part of the outcome data will be collected using telephone interviews. To minimise bias, these interviews will be conducted by a research assistant following a structured script.

### Intervention group
In the intervention group, a tailored approach to informing relatives at risk will be provided. In this approach, probands with a likely pathogenic or pathogenic variant will discuss with the genetic counsellor which relatives are at risk of inheriting the familial variant. They will then be asked which of these relatives they prefer to inform themselves at first using a family letter written by the genetic counsellor, and which relatives they prefer to be directly informed by the genetic counsellor with a similar family letter. This will be discussed during routine post-test counselling. In both cases, after 1 month, the genetic counsellor will send the family letter directly to all relatives at risk for whom the proband has provided consent to contact. The proband will be asked to provide contact details of these relatives.

The family letter is standardised for all three participating centres. For the intervention group, the letter also includes a link to a website specifically designed for this study where relatives can find additional information (www.familieleden.erfelijkehartziekten.nl). The information on this website will be tailored to relative's situations (ie, specified for disease-type, hospital, parenthood, whether relatives have a desire to have children in the future and which information relatives prefer to receive) by asking them to fill out a short questionnaire on their first visit to the website.

### Control group
In the control group, the standard care approach will be used. If a likely pathogenic or pathogenic variant is identified, probands assigned to the control group will be asked by the genetic counsellor to inform relatives at risk about the genetic test result, the consequences of this result for relatives and the advice regarding predictive DNA testing and/or cardiac monitoring. This will be discussed during routine post-test counselling. Probands will be supported in informing relatives at risk by a family letter written by the genetic counsellor. This family letter is also standardised for all three participating centres. However, this letter does not include the link to the website with tailored information described earlier, but does include a link to a general website on ICCs (www.erfelijkehartziekten.nl).

### Measurement time points
For secondary outcome measures, participating probands will be asked to complete a questionnaire 1 month after receiving the genetic test result (T1) and to complete a second questionnaire 9 months after the test result (T2). Before T1 and T2, a short structured telephone interview will be conducted about participant's knowledge of which relatives are at risk of ICCs and which relatives are informed, because these items are expected to be too complex to answer in a questionnaire.[33] Participating relatives will complete one questionnaire after attending genetic counselling.

## Measures
### Primary outcome measures
To assess the effect of a tailored approach to informing relatives at risk, the difference between the intervention and control groups in uptake of (1) genetic counselling and (2) predictive DNA testing of relatives at risk will be measured. To do this, the number of relatives attending genetic counselling and the number of relatives who are genetically tested in the first year after detection of the

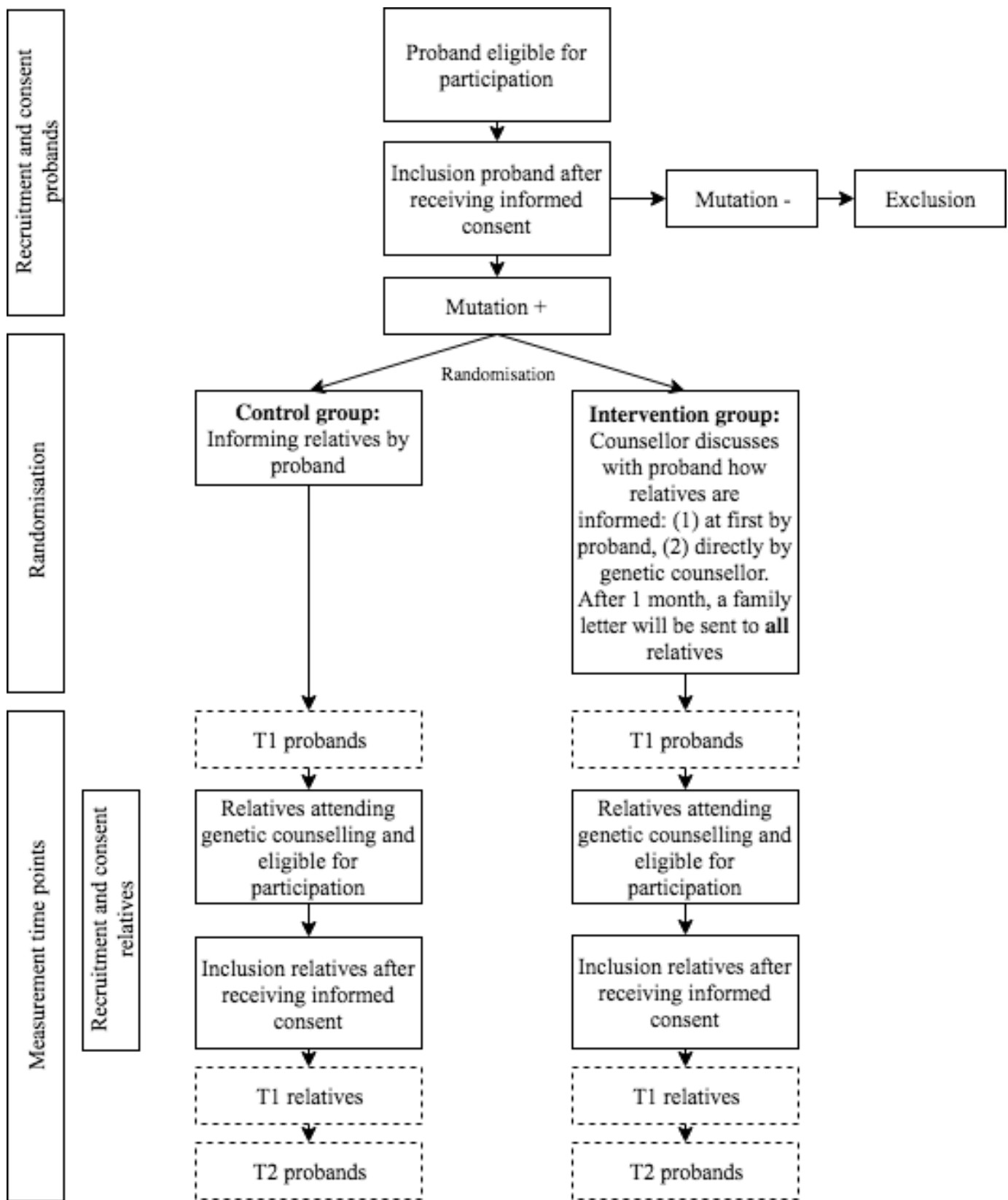

**Figure 1** Flowchart of the study procedure.

likely pathogenic or pathogenic variant in the proband will be collected in the laboratories of each participating centre. DNA test results of relatives counselled in non-participating centres will also be taken into account because, in the Netherlands, predictive DNA testing of relatives is always performed in the same laboratory where the proband was tested.

The number of relatives attending genetic counselling and undergoing predictive DNA testing will be compared with the total number of relatives at risk of inheriting the variant who are eligible for genetic counselling and predictive DNA testing based on family pedigrees. For relatives who attend genetic counselling but decide against predictive DNA testing, subsequent attendance of cardiac monitoring will be checked.

Relatives at risk who are eligible for genetic counselling and predictive DNA testing are first-degree relatives and second-degree relatives if there is a connecting deceased first-degree relative suspected of having an ICC. Following the Dutch clinical guidelines for cardiomyopathies, relatives at risk are eligible for genetic counselling and predictive DNA testing from the age of 10 years. For primary arrhythmias, depending on the specific arrhythmic disorder, relatives at risk are eligible for predictive DNA testing from birth.

Furthermore, conditional uptake of relatives at risk, defined as the number of relatives who are genetically tested relative to the number who attend genetic counselling, will be calculated. Uptake will be measured at randomisation condition (intervention or control group) and family level.

### Secondary outcome measures

Secondary outcome measures will be measured using both validated and self-constructed questionnaire items. An overview of these items is shown in the online supplementary material S2. Secondary outcome measures include the following:

*Appreciation of the information provision strategy used and preferences regarding the approach used to inform relatives at risk*: This will be evaluated in both probands and relatives using self-constructed items on a 5-point Likert scale (1 = 'Totally disagree' to 5 = 'Totally agree') in a questionnaire (probands: five items, range 5–25; relatives: six items, range 6–30). Probands will be asked to answer an additional self-constructed item during the structured telephone interview about whether they would have preferred to inform their relatives differently. Two additional self-constructed items will be administered in the intervention group to assess decisional conflict in probands, including whether probands thought it was difficult to choose to inform their relatives themselves or have them informed by the counsellor, and whether they were satisfied by their decision, on a 5-point Likert scale (1 = 'Totally disagree' to 5 = 'Totally agree'; range 2–10). Probands will be asked to fill out these items at T1. At T2, a self-constructed item will be administered to assess whether their opinion regarding the approach used has changed. The questionnaire for relatives also includes a self-constructed item on how they were informed (ie, by whom they were informed and what information was provided). Finally, probands (at T1 and T2) and relatives will be asked whether they visited the website www.erfelijkehartziekten.nl and, if yes, how they evaluated the website, using

four self-constructed items on a 5-point Likert scale (1 = 'Totally disagree' to 5 = 'Totally agree'; range 4–20).
*Impact on family communication*: To assess the impact of the tailored approach versus the usual care approach on family functioning, probands (at T1 and T2) and relatives will be asked to fill out an adapted version of the Openness to Discuss Cancer in the Family (ODCF) scale, which assesses communication about genetic risks within families with nine items on a 5-point Likert scale (1 = 'Totally disagree' to 5 = 'Totally agree'; range 9–45).[34] Psychometric characteristics of the original ODCF scale are satisfactory.[34] In addition, a self-constructed item will be administered asking about the nature of regular communication with relatives and whether probands and relatives experienced changes in their relationships with relatives as a consequence of the information provision process.
*Impact on psychological functioning*: To assess the impact on psychological functioning, two validated questionnaires will be administered in probands (at T1 and T2) and relatives. Participants will be asked to fill out an adapted version of the Cancer Worry Scale (CWS).[35] The CWS was developed and validated in Dutch patients with hereditary types of cancer.[35] Because it was validated in a Dutch patient population and is previously used in a genetic patient population, it was considered the most appropriate scale for this randomised controlled trial. The CWS consists of eight items on a 4-point Likert scale (1 = 'Almost never' to 4 = 'Almost always'; range 8–32). Psychometric characteristics of the CWS have been assessed in a sample of breast cancer survivors, and support its reliability and validity.[35]

In addition, the Hospital Anxiety and Depression Scale (HADS) will be administered to assess whether participants experience anxious or depressed feelings after being informed about the hereditary disease.[36] The HADS contains two 7-item subscales on a 4-point Likert scale with diverse answer options that assess both anxiety and depression with a score range of 0–21. Psychometric characteristics of the HADS were assessed as good.[36]

### Participants' characteristics

To assess whether randomisation succeeded and whether characteristics of participating probands and relatives have influenced the primary and secondary outcome measures, sociodemographic and clinical factors will be collected, including gender, education level, ethnicity, living situation and parenthood, family history and the diagnosis of the probands at T1. Relatives will additionally be asked what their degree of kinship is with the proband.

For the same reason, psychosocial and personality factors will be assessed in both probands (at T1) and relatives. Coping style will be assessed by using the shortened version of the Threatening Medical Situations Inventory (TMSI).[37 38] The TMSI assesses a 'monitoring' versus 'blunting' coping style related to a medical threat, and it was previously evaluated in an oncogenetic patient

population.[37 39] The shortened version of the TMSI contains two subscales, both consisting of six items on a 5-point Likert scale (1 = 'Totally not applicable' to 5 = 'Totally applicable'; range 6–30). Reliability and validity are satisfactory.[37 38] The Trait subscale of the State Trait Anxiety Inventory (STAI) will be administered to assess trait anxiety in both probands and relatives.[40] The STAI is frequently used in research settings and consists of 20 items on a 4-point Likert scale (1 = 'Not at all' to 4 = 'Very much so'; range 20–80).[40] The reliability and validity for the Dutch translation of the STAI are assessed as good.[41]

Self-efficacy and perceived motivators and barriers regarding informing relatives at risk will be assessed using an adapted version of the 'motivation' and 'self-efficacy' subscales of the Informing Relatives Inventory (IRI).[42] The IRI was developed and evaluated in an oncogenetic patient population, and showed satisfactory reliability and validity.[42] The 'motivation' subscale consists of 30 items on a 5-point Likert scale (1 = 'No role' to 5 = 'A large role'; range 30–150), and the 'self-efficacy' subscale consists of 7 items on a 4-point Likert scale (1 = 'Not sure at all' to 4 = 'Very sure'; range 7–21). Probands will also be asked to answer a self-constructed item during the telephone interviews regarding whether relatives were informed and whether probands intended to inform (remaining) at-risk relatives.

Risk perception regarding the risk of relatives carrying the variant and developing the disease will be assessed by using self-constructed items. These items ask participants to rate the perception of the risk of relatives carrying the variant and developing the disease on a scale from 1 (lowest risk) to 10 (highest risk) as well as from 0% (lowest risk) to 100% (highest risk).

Health literacy—defined as the ability to obtain, process and understand basic health information and services—will be assessed in probands and relatives using the items on the 'functional health literacy' and 'communicative health literacy' subscales of the 3HL questionnaire.[43] Both subscales contain five items on a 4-point Likert scale (1 = 'Never' to 4 = 'Often'; range per subscale 5–20). The reliability for both scales was assessed as high and the validity as satisfactory.[43]

## Sample size calculation

The study aims to detect a difference of 15% in uptake of genetic counselling by relatives between the control (usual care, 50%) and intervention groups (tailored approach, 65%). Assuming a two-sided 5% significance level and a power of 80%, 340 relatives (170 in each group) would be required to participate in this study. On average, six relatives per proband are at 50% risk of inheriting the variant, including children and adults.[9] With a conservative estimate of four eligible adult relatives per proband at risk, 85 probands with an ICC and an identified likely pathogenic or pathogenic (class 4 or 5) variant will need to be included in this study to reach 340 relatives. A likely pathogenic or pathogenic variant is found in, on average, 20% (lower margin) of all probands with

a suspected ICC. With an expected response rate of 70% and a drop-out rate of 20%, approximately 759 probands will be approached to participate in the study.

## Data analysis
### Statistical analysis

Sociodemographic, clinical, psychosocial and personality variables will be analysed using descriptive and frequency statistics. An intention-to-treat approach will be used. SPSS V24.0 will be used to perform statistical analyses.[44] An $\alpha$ level of $p<0.05$ will be used. Analysis of variance and $\chi$ tests will be used to assess differences (1) in sociodemographic, clinical and psychological characteristics between the intervention and control groups and (2) in participants and non-participants, as appropriate. Descriptive and frequency statistics will be used to describe the primary outcomes: (1) uptake of genetic counselling and (2) uptake of predictive DNA testing. Logistic regression analysis will be conducted to assess differences between the intervention and control groups on the primary outcomes, with the randomisation group as the main exploratory variable. Two logistic regression models will be used, with the first model including only the randomisation group and the second model also including the potential covariates (ie, sociodemographic, clinical and psychological variables). Multilevel analyses will be performed to assess whether the randomisation group, that is, the independent variable, has an impact on family and psychological functioning, that is, the secondary outcome variables. The two measurement time points in probands will be treated as nested within probands. To prevent influence of potential confounding factors, multilevel analysis will be adjusted for covariates as well. Participant appreciation of the approach used will be described using frequency statistics.

### Qualitative analysis

Open questions will be analysed using thematic analysis based on the principles of Braun and Clarke.[45] Analysis software for qualitative data, MAXQDA V12, will be used.[46] Two trained coders will conduct the coding analysis of open answer options independently. Codes will be discussed and modified by the two coders until agreement is met. Subsequently, the coders will analyse and interpret the codes to create a structure of main themes and subthemes. The qualitative results will be used to supplement the questionnaire data.

### Patient and public involvement

Prior to this randomised controlled trial, face-to-face interviews were conducted with probands and counselled relatives (both carriers and non-carriers) to explore their experiences with and preferences regarding informing at-risk relatives (unpublished). In addition, online focus groups were conducted with HCPs. The randomised controlled trial was then designed based on the findings of both these interviews and focus groups. Since this study is part of the eDETECT (Early Detection of

Cardiomyopathy Mutation Carriers) research consortium (CVON2015-12), several patient representative groups (the PLN Foundation; Harteraad, Heartz) participated in the user committee and scientific meetings and thereby gave input to this research proposal. Patients are not involved in the recruitment and conduct of the study.

During patient seminars, patients will be updated on the progress and results of the study. In addition, during the eDETECT scientific meetings, all participants of the eDETECT consortium (including representatives of the aforementioned patient organisations) will be informed. After completion of the study, group results will be disseminated by e-mail to study participants who indicated their interest in the outcome during informed consent. A summary of the results will also be posted on the ICC website (www.erfelijkehartziekten.nl).

The burden of the intervention was not assessed because this is an intrinsic part of the outcome measures of this study. The patients themselves were involved in pilot testing the questionnaires used to assess these outcome measures.

## ETHICS AND DISSEMINATION
Informed consent is required from each participant. Participants who provide written informed consent can withdraw from the study at any time, without providing a reason.

After receiving informed consent, a unique research ID will be assigned to the participant. Only this ID will be used to identify research documents. Each research document will be saved on a secured server. The principal investigator, coordinating investigator and executing investigator have access to this secured server. Research documents will be saved for a period of 15 years. This randomised controlled trial is registered at the Netherlands Trial Register. Separate manuscripts with findings on, respectively, the primary and secondary outcomes will be published in peer-reviewed journals.

## TRIAL STATUS
Recruitment of probands during pre-test genetic counselling for this randomised controlled trial started in November 2017. In total, recruitment of probands will last 1 year. Subsequent uptake of genetic counselling and predictive DNA testing will be measured until 1 year after the detection of a pathogenic variant in the proband. Data collection will therefore continue until January 2020, taking into account a duration of, on average, 3 months for the DNA-test result in the proband to be available. To date, 68 probands have been included and randomised to either the intervention or the control group. In addition, 49 relatives consented to participate.

**Acknowledgements** Patient advisors are acknowledged for their input regarding the design of this randomised controlled trial.

**Contributors** LMvdH and IC designed the intervention. LMvdH, IC, EMAS and JPvT were involved in conception and design of the protocol. LMvdH was involved in drafting the manuscript. IC, EMAS, JPvT, YMH and AFB critically revised the manuscript. All authors were involved in the final approval of the manuscript.

**Funding** We acknowledge the financial support from the Netherlands Cardiovascular Research Initiative: an initiative with support of the Dutch Heart Foundation, CVON2015-12 eDETECT and CVON2017-10 DOLPHIN-GENESIS.

**Competing interests** None declared.

**Patient consent for publication** Not required.

**Ethics approval** Ethics approval for this work was provided by the Medical Research Ethics Committee Academic Medical Centre (MEC 2017-145).

**Provenance and peer review** Not commissioned; externally peer reviewed.

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
