## [Reviewer comments · BMJ Open]

ARTICLE DETAILS

TITLE (PROVISIONAL)	A tailored approach towards informing relatives at risk of inherited cardiac conditions: study protocol for a randomised controlled trial
AUTHORS	van den Heuvel, Lieke; Hoedemaekers, Yvonne; Baas, Annette; van Tintelen, J; Smets, Ellen; Christiaans, Imke

VERSION 1 - REVIEW

REVIEWER	Cynthia A. James Johns Hopkins University, Division of Cardiology Baltimore, Maryland, USA
REVIEW RETURNED	20-Aug-2018

GENERAL COMMENTS	This is a study protocol for a randomized 3-center study of a tailored approach vs. usual care to inform relatives of genetic risk following detection of a proband with a pathogenic or likely pathogenic variant associated with an autosomal dominant inherited cardiovascular disease. Identifying effective approaches to family communication in this situation is important. Professional recommendations increasingly recommend genetic testing of probands affected by these cardiac conditions in large part to facilitate cascade genetic testing. However, as the authors nicely summarize uptake of genetic screening by relatives has been relatively modest even in countries with nationalized healthcare systems with little required out-of-pocket cost associated with the cascade testing and necessary follow-up cardiovascular screening. One documented barrier has been communication of information about genetic risk and the process of genetic testing from probands to relatives. This very nice study develops a multipronged approach (tailored options for informing relatives, access to a website with personalized information, and possibly an additional discussion with a genetic counselor about identifying at-risk relatives) and compares it to usual care which includes a letter given to probands to pass along to relatives. The primary outcome is an increase in uptake of genetic counseling and predictive DNA testing. Secondary outcomes are related to family and individual psychosocial function as well as appreciation of and attitudes toward the process of informing relatives. Strengths of this study include strong collaboration evident between the genetics / genetic counseling team and psychologists with requisite expertise. Additional strengths include a multicenter approach and a critical clinical question about which there is limited data currently to inform practice.
--

The primary weakness is the fact that the investigators are using multiple interventions simultaneously without further stratification – probably to prevent loss of power. The downside of this approach is that ultimately while the investigators will be able to address the question: “If we make a substantial effort including a personalized website, opportunities for tailored communication to relatives including getting a letter directly from the genetic counselor, and extra conversation about family communication between the genetic counselor and proband can the uptake of cascade genetic testing in familial cardiovascular disease be increased?” This is an undeniably important question and certainly if the answer is “no” it rules out any of these approaches. However, if cascade testing is increased, the authors will be unable to distinguish which aspect of their approach was most (or least) important.

There are no major issues of concern.

Minor issues

1. In the introduction it would be helpful for the authors to briefly discuss other prior or ongoing trials of methods for increasing uptake of cascade testing of relatives. In particular a study is underway for families with dilated cardiomyopathy which is directly relevant: Kinnamon DD et al, PMID 29237686. Another potentially relevant reference is Hodgson et al; PMID: 26130486.

2. Could the authors clarify whether the intervention includes an additional conversation between the genetic counselor and the proband focused on family communication in which the proband and genetic counselor discuss which relatives are at risk and makes a plan for contact for each? Does this extra conversation also happen in the control arm – but without the offer for direct contact – or is this discussion limited to what occurs in the genetic counseling session?

3. It would be nice to know enrollment to date in the “trial status” section (page 17).

4. If possible it would be helpful to include a model consent form as Supplementary Material as suggested in the SPIRIT checklist.

5. While I recognize that no changes to design are possible, I'd encourage the authors in the future to consider using the Cardiac Anxiety Questionnaire (Eifert et al, PMID 11004742) as a measure of impact on psychological function. It is validated in cardiovascular and general populations and may be a particularly helpful addition to or in place of the revised cancer worry scale used.

6. Finally there are a few small grammar / word choice errors:

a. Page 8, line 37: “neither participants nor the genetic counsellors will and can be blinded”. Should be will OR can.

b. Page 9, line 27: “The information on the website is tailored to the relatives’ situation... whether they have a child wish and/or children...” This probably is asking whether the relative wants children someday or more likely whether they relatively is actively planning a pregnancy.

c. Page 10, line 34 and page 11, line 18 – “the number of relatives attending genetic counselling as well as the number of relatives

	that is genetically tested in the first year...”. This should be the number of relatives WHO ARE genetically tested
--	---

REVIEWER	Laura Forrest Peter MacCallum Cancer Centre Australia
REVIEW RETURNED	11-Oct-2018

GENERAL COMMENTS	Strength: This is not the first RCT to evaluate a tailored intervention to support family communication of genetic information (see Hodgson et al. 2014 BMC Med Genet 15: 33) Limitation: It is unclear why it is a limitation that only relatives of probands included in the study can be invited to participate. It would be unethical and illogical to invite relatives of probands not included in this RCT. Introduction offers a clear summary of the topic with most of the relevant and up to date literature cited. There has been a study in a clinical setting (rather than a research setting, as noted in text), where genetic counsellors provided more support to probands informing at-risk relatives. This study demonstrated a significant increase in uptake of genetic counselling and predictive genetic testing (see Forrest et al. 2008 Genet Med 10(3):167). Further, and as noted above responding to the strengths listed, Hodgson et al. (2014) have published an RCT based in a clinical genetics setting providing support to probands in the intervention arm to communicate to their at-risk relatives, thereby preserving their autonomy. There are no limitations described for this study. The standard of written English is mostly fine throughout the paper, however, there are quite a few places in the text and supplementary material where there are grammatical errors.
--

REVIEWER	Yvonne Bombard. My graduate student Chloe Mighton assisted with this review Li Ka Shing Knowledge Institute, St. Michael's Hospital, Canada
REVIEW RETURNED	13-Oct-2018

GENERAL COMMENTS	Thank you for the opportunity to review this interesting article that describes the protocol for an RCT testing a novel, tailored approach to address familial risk communication about hereditary heart conditions.  1. The primary outcome actually consists of two outcomes – number of relatives who have genetic counseling, and number of relatives who have genetic testing. Will these outcomes be considered separately, or does the relative need to have both counseling and testing for the outcome to be counted? For instance, how would the authors count a relative who had counseling and testing, versus a relative who only had testing? Greater clarity about how this outcome is measured is needed. 2. A potential confounder for the primary outcome is the proband's baseline intent to communicate their genetic test results to relatives. Other confounders could be whether individuals are even
---

	in contact with their relatives, or the nature of their regular communication with their relatives. Have the authors considered capturing this data and including it in their analysis? If not, these should be stated as potential confounders and limitations. 3. There are other variables that may confound the relationship between intervention and the primary outcome. These could include but are not limited to, sociodemographic characteristics, baseline intent to share results with relatives, and psychological characteristics. The authors could consider assessing the relationship between their primary outcome and potential confounding variables in an exploratory analysis. Otherwise, the possibility of other variables confounding the relationship between the intervention and the primary outcome could be listed as a limitation. 4. Will all eligible probands who are seen at the recruiting clinics be approached to participate? If not, there is a potential for sampling bias. 5. Have the authors considered administering the psychological measures at baseline, before randomization? Baseline scores could then be adjusted for in the regression analyses that the authors describe, as baseline psychological state may be a confounder. 6. The authors state that open ended questions will be analysed though thematic analysis. More detail about how the authors plan to conduct this thematic analysis, and any relevant references, should be included. The authors should also state how they intend to triangulate or analyze their mixed-methods results, if at all.
--	---

VERSION 1 – AUTHOR RESPONSE

Reviewer(s)' Comments to Author:

Reviewer: 1

Reviewer Name: Cynthia A. James

Institution and Country: Johns Hopkins University, Division of Cardiology Baltimore, Maryland, USA

Please state any competing interests or state 'None declared': None declared.

Please leave your comments for the authors below General Comments:

This is a study protocol for a randomized 3-center study of a tailored approach vs. usual care to inform relatives of genetic risk following detection of a proband with a pathogenic or likely pathogenic variant associated with an autosomal dominant inherited cardiovascular disease. Identifying effective approaches to family communication in this situation is important. Professional recommendations increasingly recommend genetic testing of probands affected by these cardiac conditions in large part to facilitate cascade genetic testing. However, as the authors nicely summarize uptake of genetic

screening by relatives has been relatively modest even in countries with nationalized healthcare systems with little required out-of-pocket cost associated with the cascade testing and necessary follow-up cardiovascular screening. One documented barrier has been communication of information about genetic risk and the process of genetic testing from probands to relatives. This very nice study develops a multipronged approach (tailored options for informing relatives, access to a website with personalized information, and possibly an additional discussion with a genetic counselor about identifying at-risk relatives) and compares it to usual care which includes a letter given to probands to pass along to relatives. The primary outcome is an increase in uptake of genetic counseling and predictive DNA testing. Secondary outcomes are related to family and individual psychosocial function as well as appreciation of and attitudes toward the process of informing relatives.

Strengths of this study include strong collaboration evident between the genetics / genetic counseling team and psychologists with requisite expertise. Additional strengths include a multicenter approach and a critical clinical question about which there is limited data currently to inform practice.

The primary weakness is the fact that the investigators are using multiple interventions simultaneously without further stratification – probably to prevent loss of power. The downside of this approach is that ultimately while the investigators will be able to address the question: “If we make a substantial effort including a personalized website, opportunities for tailored communication to relatives including getting a letter directly from the genetic counselor, and extra conversation about family communication between the genetic counselor and proband can the uptake of cascade genetic testing in familial cardiovascular disease be increased?” This is an undeniably important question and certainly if the answer is “no” it rules out any of these approaches. However, if cascade testing is increased, the authors will be unable to distinguish which aspect of their approach was most (or least) important.

Authors’ response: We thank the reviewer for this comment. We are aware of this limitation of our randomised controlled trial. We therefore added this as a limitation in the 'Strengths and limitations of this study' section.

Page 3, lines 9-10: “In this trial, evaluation of the effect on outcome of different components of the intervention is not possible, due to limited power.”

There are no major issues of concern.

Minor issues

1. In the introduction it would be helpful for the authors to briefly discuss other prior or ongoing trials of methods for increasing uptake of cascade testing of relatives. In particular a study is underway for families with dilated cardiomyopathy which is directly relevant: Kinnamon DD et al, PMID 29237686. Another potentially relevant reference is Hodgson et al; PMID: 26130486.

Authors’ response: We thank the reviewer for this helpful comment. We added a short section about prior and ongoing trials for improving cascade testing of relatives in the

'introduction' section:

Page 5, lines 8-11: "Previous studies assessing interventions to enhance family communication in hereditary diseases showed that some interventions are effective in increasing the uptake of genetic [19-21]. An intervention trial aimed at improving family communication in specifically dilated cardiomyopathy is still ongoing [22]."

2. Could the authors clarify whether the intervention includes an additional conversation between the genetic counselor and the proband focused on family communication in which the proband and genetic counselor discuss which relatives are at risk and makes a plan for contact for each? Does this extra conversation also happen in the control arm – but without the offer for direct contact – or is this discussion limited to what occurs in the genetic counseling session?

Authors' response: We thank the reviewer for this comment. There is no additional conversation between the proband and the genetic counsellor on family communication included in the intervention arm. In both the intervention and the control arm, family communication is discussed during routine pre- and post-test counselling. In the intervention group, genetic counsellors discuss additionally which relatives probands prefer to inform themselves and which relatives they prefer to be informed by the genetic counsellor. To clarify this in the manuscript, a sentence in the methods section describing the intervention and the control condition has been added:

Page 9, lines 3-5: "... and which relatives they prefer to be directly informed by the genetic counsellor with a similar family letter. This will be discussed during routine post-test counselling."

Page 9, line 17-21: "If a likely pathogenic or pathogenic variant is identified, probands assigned to the control group will be asked by the genetic counsellor to inform relatives at risk about the genetic test result, the consequences of this result for relatives and the advice regarding predictive DNA testing and/or cardiac monitoring. This will be discussed during routine post-test counselling."

3. It would be nice to know enrollment to date in the "trial status" section (page 17).

Authors' response: We thank the reviewer for this helpful comment. The enrollment to date has been added to the 'trial status' section:

Page 18, lines 1-3: "To date, 68 probands have been included and randomised to either the intervention or the control group. In addition, 49 relatives consented to participate."

4. If possible it would be helpful to include a model consent form as Supplementary Material as suggested in the SPIRIT checklist.

Authors' response: We thank the reviewer for this comment. We understand that a model consent form for probands and relatives would be helpful to include in this study protocol. These were added to the Supplementary Material (Supplementary Material S3).

5. While I recognize that no changes to design are possible, I'd encourage the authors in the future to consider using the Cardiac Anxiety Questionnaire (Eifert et al, PMID 11004742) as a

measure of impact on psychological function. It is validated in cardiovascular and general populations and may be a particularly helpful addition to or in place of the revised cancer worry scale used.

Authors' response: We understand the comment of the reviewer and want to thank her for this suggestion. We decided to use a revised version of the Cancer Worry Scale, because this scale is developed and validated in a Dutch patient population and previously evaluated in a genetic patient population. The Cardiac Anxiety Questionnaire has been validated in the Netherlands, but unfortunately only in patients with acute coronary syndrome and not in patients with inherited (cardiac) diseases. We agree with the reviewer that the Cardiac Anxiety Questionnaire would have been a helpful addition. We sincerely hope the reviewer respects our choice. To explain this, a sentence was added to the 'Measures' section:

Page 12, lines 20-24: "The CWS was developed and validated in Dutch patients with hereditary types of cancer [35]. Because the CWS is validated in a Dutch patient population and is previously used in a genetic patient population, it was considered the most appropriate scale for this randomised controlled trial. The CWS consists of eight items on a 4 point Likert scale (i.e., 1= 'Almost never' to 4 = 'Almost always'; range 8-32)."

6. Finally there are a few small grammar / word choice errors:

a. Page 8, line 37: "neither participants nor the genetic counsellors will and can be blinded". Should be will OR can.

b. Page 9, line 27: "The information on the website is tailored to the relatives' situation... whether they have a child wish and/or children..." This probably is asking whether the relative wants children someday or more likely whether they relatively is actively planning a pregnancy.

c. Page 10, line 34 and page 11, line 18 – "the number of relatives attending genetic counselling as well as the number of relatives that is genetically tested in the first year...". This should be the number of relatives WHO ARE genetically tested

Authors' response: We thank the reviewer for these helpful adjustments. These grammar/word choice errors are corrected in the manuscript. In addition, a native English editor has revised the manuscript.

Page 8, line 17: "Neither participants nor the genetic counsellors will or can be blinded for group assignment.."

Page 9, line 10-14: "The information on this website is tailored to the relatives' situation (i.e., specified for disease type, hospital, parenthood, whether relatives have a desire to have children in the future, and which information relatives prefer to receive) by asking them to fill out a short questionnaire on their first visit to the website."

Page 10, lines 14-16: "To do this, the number of relatives attending genetic counselling and the number of relatives who are genetically tested in the first year..."

Page 11, line 7-8: "Furthermore, conditional uptake of relatives at risk, defined as the number of relatives who are genetically tested..."

Reviewer: 2

Reviewer Name: Laura Forrest

Institution and Country: Peter MacCallum Cancer Centre Australia

Please state any competing interests or state 'None declared': None declared

Please leave your comments for the authors below

Strength: This is not the first RCT to evaluate a tailored intervention to support family communication of genetic information (see Hodgson et al. 2014 BMC Med Genet 15: 33)

Authors' response: We thank the reviewer for this helpful comment. We adjusted the 'strengths and limitation' section, in accordance to the editor's comments. This strength was therefore replaced, as shown in the first page of this document as well. We have added the study of Hodgson et al (2014/2016) to the 'introduction' section.

Page 3, lines 6-7: "This study will be conducted in three clinical genetic clinics with expertise on cardiogenetics, which will facilitate participant inclusion."

Page 5, lines 8-10: "Previous studies assessing interventions to enhance family communication in hereditary diseases showed that some interventions are effective in increasing the uptake of genetic counselling [19-21]."

Limitation: It is unclear why it is a limitation that only relatives of probands included in the study can be invited to participate. It would be unethical and illogical to invite relatives of probands not included in this RCT.

Authors' response: We agree with the reviewer that it would be inappropriate to invite relatives of probands not included in this RCT. However, we meant that it is a limitation that relatives of participating probands, who do not attend genetic counselling, cannot be approached for study participation and cannot give their opinion on the used approach to inform them. Unfortunately, this may induce a bias, because relatives having a more positive attitude towards being informed about the inherited cardiac disease diagnosed in their family will be more likely to attend genetic counselling. These relatives will possibly also have a more positive attitude towards the approach used to inform them. To avoid confusion, this limitation has been replaced in the 'Strengths and limitations of this study' section.

Introduction offers a clear summary of the topic with most of the relevant and up to date literature cited. There has been a study in a clinical setting (rather than a research setting, as noted in text), where genetic counsellors provided more support to probands informing at-risk relatives. This study demonstrated a significant increase in uptake of genetic counselling and predictive genetic testing (see Forrest et al. 2008 Genet Med 10(3):167). Further, and as noted above responding to the strengths listed, Hodgson et al. (2014) have published an RCT based in a clinical genetics setting providing support to probands in the intervention arm to communicate to their at-risk relatives, thereby preserving their autonomy.

Authors' response: We thank the reviewer for this comment. The studies we referred to were studies evaluating a direct contact approach in a research setting, rather than additional counselling.

However, We agree with the reviewer that the references mentioned (i.e., Forrest et al, 2008; Hodgson, et al., 2014/2016) are useful additions to the manuscript.

Therefore, we have added the references to the 'introduction' section (see response to comment 1 of reviewer 1).

There are no limitations described for this study.

Authors' response: We thank the reviewer for this comment. The most important limitations are however described in the 'Strengths and limitations of this study' section. Because the format of the BMJ open does not require a specific limitations section and our manuscript already reached the maximum word count, we decided to only shortly describe the most important limitations in the "strengths and limitations of this study" section after the abstract (see also editor comments).

The standard of written English is mostly fine throughout the paper, however, there are quite a few places in the text and supplementary material where there are grammatical errors.

Authors' response: We thank the reviewer for this comment. A native English editor has revised the manuscript.

Reviewer: 3

Reviewer Name: Yvonne Bombard. My graduate student Chloe Mighton assisted with this review

Institution and Country: Li Ka Shing Knowledge Institute, St. Michael's Hospital, Canada

Please state any competing interests or state 'None declared': None declared

Please leave your comments for the authors below

Thank you for the opportunity to review this interesting article that describes the protocol for an RCT testing a novel, tailored approach to address familial risk communication about hereditary heart conditions.

1. The primary outcome actually consists of two outcomes – number of relatives who have genetic counseling, and number of relatives who have genetic testing. Will these outcomes be considered separately, or does the relative need to have both counseling and testing for the outcome to be counted? For instance, how would the authors count a relative who had counseling and testing, versus a relative who only had testing? Greater clarity about how this outcome is measured is needed.

Authors' response: We agree with the reviewer that the primary outcome indeed consists of two outcomes: (1) The number of relatives who attend genetic counselling, (2) The number of relatives who decide to have predictive genetic testing. These outcomes will be analysed and reported separately. We have changed the 'measures' and 'data analysis' sections to clarify this.

Page 2, lines 14-15: "Primary outcomes are uptake of genetic counselling and predictive DNA testing in relatives (total sample n = 340 relatives). Secondary outcomes are appreciation of the used approach and impact on family- and psychological functioning, which will be assessed using questionnaires.

Page 10, lines 11-14: "Primary outcome measures - To assess the effect of a tailored approach towards informing relatives at risk, the difference between the intervention- and control group in uptake of (1) genetic counselling, and (2) predictive DNA testing of relatives at risk will be measured."

Page 15, lines 17-19: "Descriptive and frequency statistics will be used to describe the primary outcomes: (1) uptake of genetic counselling, and (2) uptake of predictive DNA testing."

2. A potential confounder for the primary outcome is the proband's baseline intent to communicate their genetic test results to relatives. Other confounders could be whether individuals are even in contact with their relatives, or the nature of their regular communication with their relatives. Have the authors considered capturing this data and including it in their analysis? If not, these should be stated as potential confounders and limitations.

Authors' response: We thank the reviewer for this valuable suggestion. Self-constructed items are administered during the telephone interviews regarding whether relatives are informed and whether probands intend to inform (remaining) at-risk relatives. Furthermore, participants are asked to indicate whether their relationship with relatives changed. The

'measures' section was changed to clarify this.

Unfortunately, due to the design of the study, it was not possible to include a baseline measure prior to randomisation to assess potential confounders, such as this one. All probands attending pre-test genetic counselling are informed about the study and asked for participation prior to receiving the genetic test result. Probands are definitively included and randomised when a likely pathogenic or pathogenic genetic variant is identified. Directly after randomisation, disclosure of the test result (i.e. post-test genetic counselling) takes place, in which informing relatives, dependent on randomisation, is discussed. Analysis will be adjusted for covariates that are collected. The 'data analysis' section has now been revised accordingly. In addition, we added a sentence to the 'Strengths and limitations of this study' section.

Page 12, lines 13-16: "In addition, a self-constructed item will be administered asking about the nature of regular communication with relatives and whether probands and relatives experienced changes in their relationships with relatives as a consequence of the information provision process."

Page 14, lines 7-9: "Probands will also be asked to answer a self-constructed item during the telephone interviews regarding whether relatives were informed and whether probands intended to inform (remaining) at-risk relatives."

Page 3, line 12-14: "Because a baseline measure for the secondary outcomes is not possible, we cannot control for likely confounders such as intention to inform at-risk relatives, and family and psychological functioning at baseline."

Page 15, lines 23-24: "To prevent influence of potential confounding factors, analysis will be adjusted for covariates (i.e., sociodemographic, clinical and psychological variables)."

3. There are other variables that may confound the relationship between intervention and the primary outcome. These could include but are not limited to, sociodemographic characteristics,

baseline intent to share results with relatives, and psychological characteristics. The authors could consider assessing the relationship between their primary outcome and potential confounding variables in an exploratory analysis. Otherwise, the possibility of other variables confounding the relationship between the intervention and the primary outcome could be listed as a limitation.

Authors' response: We thank the reviewer for this helpful comment. We indeed are planning to control for potential confounding factors, including sociodemographic and personal/psychological characteristics. We have added this to the 'data analysis' section.

Page 15, line 19 – page 16, line 2: "Logistic regression analysis will be conducted to assess differences between the intervention- and control group on the primary outcomes. Multilevel analyses will be performed to assess whether the intervention has an impact on family and psychological functioning. The two measurement time-points will be treated as nested within probands. To prevent influence of potential confounding factors, analysis will be adjusted for controlled covariates (i.e., sociodemographic, clinical and psychological variables). Regression analyses will be conducted as well to assess the influence on the primary and secondary outcomes of sociodemographic, clinical, psychological and personality characteristics. Appreciation of the used approach will be described by using frequency statistics."

4. Will all eligible probands who are seen at the recruiting clinics be approached to participate? If not, there is a potential for sampling bias.

Authors' response: We thank the reviewer for this comment. All eligible probands are indeed approached to participate by the genetic counsellor during pre-test counselling. To clarify this, the sentence in the 'recruitment' section of the manuscript has been changed into:

Page 7, lines 6-10: "All probands aged 18 years or older with an ICC or suspicion thereof, attending pre-test genetic counselling at the cardiogenetics outpatient clinics during the inclusion period will be asked to participate if they: (1) are the first of their family to visit the cardiogenetic outpatient clinic for counselling about genetic testing for ICCs; (2) they have at least one alive adult relative; and (3) are able to read and write Dutch."

5. Have the authors considered administering the psychological measures at baseline, before randomization? Baseline scores could then be adjusted for in the regression analyses that the authors describe, as baseline psychological state may be a confounder.

Authors' response: We thank the reviewer for this comment. As described in the response to the second comment of the third reviewer, a baseline measurement – prior to randomisation - was unfortunately not possible. Therefore, baseline psychological state cannot be administered. In the manuscript, a sentence describing this limitation is added to the 'Strengths and limitations of this study' section.

Page 3, lines 12-14: "Because a baseline measure for the secondary outcomes is not possible, we cannot control for likely confounding factors, such as intention to inform at-risk relatives, and family and psychological functioning at baseline."

6. The authors state that open ended questions will be analysed though thematic analysis. More detail about how the authors plan to conduct this thematic analysis, and any relevant references,

should be included. The authors should also state how they intend to triangulate or analyze their mixed-methods results, if at all.

Authors' response: We thank the reviewer for this comment. The thematic analysis approach is added to the 'data analysis' section.

Page 16, lines 4-8: "Qualitative analysis - Open questions will be analysed using thematic analysis, based on the principles of Braun and Clarke [45]. Analysis software for qualitative data, MAXQDA version 12, will be used [46]. Coding will be conducted by two trained coders independently. The codes will be analysed and interpreted to create a structure of themes and subthemes. Qualitative results will be used to supplement the questionnaire data."

VERSION 2 – REVIEW

REVIEWER	Cynthia James Johns Hopkins University Baltimore, Maryland United States of America
REVIEW RETURNED	04-Jan-2019

GENERAL COMMENTS	Minor issues  1. There is a typo in edits to the introduction related to ongoing and recent clinical trials of cascade genetic testing – specifically one or more words is missing prior to the references. "Previous studies assessing interventions to enhance family communication in hereditary diseases showed that some interventions are effective in increasing the uptake of genetic [19-21]. 2. While the authors indicate they are adding a model consent form to the Supplementary Material – which is very helpful – this has not been uploaded on the website. Can the authors make sure this happens? Thanks!
---

REVIEWER	Yvonne Bombard. My graduate student Salma Shickh assisted with this review. Li Ka Shing Knowledge Institute, St. Michael's Hospital, Canada.
REVIEW RETURNED	16-Jan-2019

GENERAL COMMENTS	Thank you for the opportunity to re-review this article that describes the protocol for an RCT evaluating a tailored approach for communicating genetic risk for inherited cardiac conditions. Overall, the authors addressed our major concerns. They provided clarification on their primary outcome. They revised their analysis plans to account for some confounders and provided a statement in the limitations section for confounders that the study could not account for. Although they added some details about the analysis plan for the open-ended questions, we suggest that they provide a brief explanation as to how coding of the data between the two independent coders will be merged (e.g. Will they discuss their codes together?).
--

	We have some minor suggestions for the authors. Given that many probands will likely have multiple relatives who may learn about the mutation, it is possible that the relatives who come in for genetic counselling or testing may have been informed by relatives and this is what motivated them to come in for testing (as opposed to the intervention). Will relatives who come in for counselling or testing be asked who notified them and the reason they came in? It would also be helpful if the authors clarify the outcomes and the variables in their multivariable analysis plan.
--	--

VERSION 2 – AUTHOR RESPONSE

Response to comments by reviewer 1

1. There is a typo in edits to the introduction related to ongoing and recent clinical trials of cascade genetic testing – specifically one or more words is missing prior to the references. “Previous studies assessing interventions to enhance family communication in hereditary diseases showed that some interventions are effective in increasing the uptake of genetic [19-21].

Authors’ response: We thank the reviewer for this comment. The reviewer refers to the sentence: ‘Previous studies assessing interventions to enhance family communication in hereditary diseases showed that some interventions are effective in increasing the uptake of genetic counselling [19-21].’ The word ‘counselling’ was missing in our response to the first comments of the reviewers, but not in the manuscript itself.

2. While the authors indicate they are adding a model consent form to the

Supplementary Material – which is very helpful – this has not been uploaded on the website. Can the authors make sure this happens? Thanks!

Authors’ response: A model information letter and informed consent form were already added to the Supplementary Material. Unfortunately, we do not know why this was not available for the reviewer. Therefore we will re-upload the document to make sure that the reviewers and the potential readers have the model information letter and informed consent form available.

Response to comments by reviewer 3

1. Although they added some details about the analysis plan for the open-ended questions, we suggest that they provide a brief explanation as to how coding of the data between the two independent coders will be merged (e.g. Will they discuss their codes together?).

Authors’ response: We thank the reviewer for this helpful comment. We added a sentence to the ‘qualitative analysis’ section of the methods to clarify our qualitative analysis plan.

Page 16, lines 11-16: “Open questions will be analysed using thematic analysis based on the principles of Braun and Clarke [45]. Analysis software for qualitative data, MAXQDA version 12, will be used [46]. Two trained coders will conduct the coding analysis of open answer options independently. Codes will be discussed and modified by the two coders until agreement is met.

Subsequently, the coders will analyse and interpret the codes to create a structure of main themes and subthemes. The qualitative results will be used to supplement the questionnaire data.”

2. Given that many probands will likely have multiple relatives who may learn about the mutation, it is possible that the relatives who come in for genetic counselling or testing may have been informed by relatives and this is what motivated them to come in for testing (as opposed to the intervention). Will relatives who come in for counselling or testing be asked who notified them and the reason they came in?

Authors’ response: We are indeed asking relatives who attend genetic counselling to fill out a questionnaire. An item on how relatives are informed (by whom, and which information was provided) is included in this questionnaire. We added this to the methods’ section and to Supplemental Material S2. Unfortunately, we did not include an item on the reason why relatives attended genetic counselling.

Page 12, lines 3-7: “Probands will be asked to fill out these items at T1. At T2, a selfconstructed item will be administered to assess whether their opinion regarding the approach used has changed. The questionnaire for relatives also includes a self-constructed item on how they were informed (i.e., by whom they were informed and what information was provided).”

Supplementary Material S2:

1. How were you informed about the hereditary predisposition in your family?
 - a. With an information letter from the hospital, received from a relative
 - b. With a letter written by a relative
 - c. In person by a relative
 - d. With an information letter from the hospital, received from a genetic counsellor/clinical geneticist
 - e. In person by a genetic counsellor/clinical geneticist
2. What information did you receive (multiple answers are possible)?
 - a. The risk to be a carrier of the hereditary predisposition for the inherited cardiac disease in my family
 - b. The possibility to make an appointment for predictive DNA testing at an outpatient clinic Clinical Genetics
 - c. The advice to be regularly monitored by a cardiologist in the hospital
 - d. Something else, namely_____

3. It would also be helpful if the authors clarify the outcomes and the variables in their multivariable analysis plan.

Authors’ response: We would like to thank the reviewer for this comment to clarify our analyses. We added the outcomes and the independent variables to the data analysis section to clarify this section so that the text (Page 15, lines 21-25 to page 16, lines 1-7) now reads:

“Descriptive and frequency statistics will be used to describe the primary outcomes: (1) uptake of genetic counselling and (2) uptake of predictive DNA testing. Logistic regression analysis will be conducted to assess differences between the intervention- and control group on the primary outcomes, with the randomisation group as the main exploratory variable. Two logistic regression models will be used, with the first model including only the exploratory variable, and the second model also including the potential covariates (i.e., sociodemographic, clinical and psychological variables). Multilevel analyses will be performed to assess whether the randomisation group, i.e., the independent variable, has an impact on family and psychological functioning, i.e., the secondary outcome variables. The two measurement time-points in probands will be treated as nested within probands. To prevent influence of potential confounding factors, multilevel analysis will be adjusted for covariates as well.”

We hope that these additions sufficiently explains our data analysis plan.

VERSION 3 - REVIEW

REVIEWER	Yvonne Bombard. My graduate student Salma Shickh assisted with this review. Li Ka Shing Knowledge Institute, St. Michael's Hospital.
REVIEW RETURNED	08-Mar-2019

GENERAL COMMENTS	Thank you for the opportunity to re-review this article that describes the protocol for an RCT evaluating a tailored approach for communicating genetic risk for inherited cardiac conditions. Overall, the authors addressed our remaining concerns. They provided sufficient details of their analysis plan for the qualitative data and of the outcomes and variables to be used in the multivariable analysis. Furthermore, they provided details on how they would determine if relatives are informed by the proband or their intervention.
---